# Differential Expression of Amanitin Biosynthetic Genes and Novel Cyclic Peptides in *Amanita molliuscula*

**DOI:** 10.3390/jof7050384

**Published:** 2021-05-14

**Authors:** Yunjiao Lüli, Shengwen Zhou, Xuan Li, Zuohong Chen, Zhuliang Yang, Hong Luo

**Affiliations:** 1Yunnan Key Laboratory for Fungal Diversity and Green Development, Kunming Institute of Botany, Chinese Academy of Sciences, Kunming 650201, China; lvliyunjiao@mail.kib.ac.cn (Y.L.); zhoushengwen@mail.kib.ac.cn (S.Z.); fungi@mail.kib.ac.cn (Z.Y.); 2CAS Key Laboratory for Plant Diversity and Biogeography of East Asia, Kunming Institute of Botany, Chinese Academy of Sciences, Kunming 650201, China; 3University of Chinese Academy of Sciences, Beijing 100049, China; 4School of Life Sciences, Yunnan University, Kunming 650091, China; 5Department of Environmental Science and Engineering, Kunming University of Science and Technology, Kunming 650091, China; x.li@hotmail.com; 6College of Life Science, Hunan Normal University, Changsha 410081, China; chenzuohong@263.net

**Keywords:** poisonous mushroom, genome, transcriptome, MSDIN family, MS

## Abstract

*Amanita molliuscula* is a basal species of lethal *Amanita* and intrigues the field because it does not produce discernable α-amanitin when inspected by High Performance Liquid Chromatography (HPLC), which sets it apart from all known amanitin-producing (lethal) *Amanita* species. In order to study the underlining genetic basis of the phenotype, we sequenced this species through PacBio and Illumina RNA-Seq platforms. In total, 17 genes of the “*MSDIN*” family (named after the first five amino acid residues of the precursor peptides) were found in the genome and 11 of them were expressed at the transcription level. The expression pattern was not even but in a differential fashion: two of the *MSDIN*s were highly expressed (FPKM value > 100), while the majority were expressed at low levels (FPKM value < 1). Prolyl oligopeptidease B (POPB) is the key enzyme in the amanitin biosynthetic pathway, and high expression of this enzyme was also discovered (FPKM value > 100). The two *MSDIN*s with highest transcription further translated into two novel cyclic peptides, the structure of which is distinctive from all known cyclic peptides. The result illustrates the correlation between the expression and the final peptide products. In contrast to previous HPLC result, the genome of *A. molliuscula* harbors α-amanitin genes (three copies), but the product was in trace amount indicated by MS. Overall, transcription of *MSDIN*s encoding major toxins (α-amanitin, β-amanitin, phallacidin and phalloidin) were low, showing that these toxins were not actively synthesized at the stage. Collectively, our results indicate that the amanitin biosynthetic pathway is highly active at the mature fruiting body stage in *A. molliuscula*, and due to the differential expression of *MSDIN* genes, the pathway produces only a few cyclic peptides at the time.

## 1. Introduction

The lethal doses of amanitins, members of the amatoxin group of bicyclic octapeptides, are extremely low, and a mature fruiting body of an amanitin-producing mushroom can cause death of an adult human being [1]. According to statistics from the China Center for Disease Control and Prevention (China CDC) from 2010 to 2020, there were 8007 poisoning incidents caused by poisonous mushrooms in China with 733 deaths [2,3,4]. Among them, the amanitin-containing mushrooms caused the majority (68.0–90.9%) of those deaths. *Amanita* species mainly include two types of cyclic peptides: amatoxins and phallotoxins [1,5]. Amatoxins are highly effective inhibitors of RNA polymerase II in eukaryotes, while phallotoxins irreversibly bind to F-actin filaments in the cytoskeleton, interrupting the dynamic process of polymerization and depolymerization [6,7]. Despite the toxicity, amanitin–antibody conjugate has been considered as a potential cancer treatment [8,9]. The biosynthesis of amatoxins is mainly completed by two important groups of genes: those in the “*MSDIN*” family, which encode the precursors of the cyclic peptides; and prolyl oligopeptidase B (*POPB*), which specifically recognizes the proproteins, and cleaves and macrocyclizes them into unmodified cyclic peptides [10,11,12].

Amatoxins, phallotoxins and other related cyclic peptides are distributed in several genera of different families in the order Agaricales, i.e., *Amanita*, *Lepiota* and *Galerina*. The distribution of amanitin biosynthetic genes in the three genera is a result of horizontal gene transfer [13,14]. In *Amanita*, there are several shared major toxins, i.e., α-amanitin, β-amanitin, phalloidin and phallacidin [1,5], while *Galerina* and *Lepiota* have primarily α-amanitin (with small amounts of β-amanitin in a few cases) [14,15]. *Amanita* is probably the last genus to obtain the ability for the toxin production, but clearly it has advanced the capability by producing diversified cyclic peptides [13].

Lethal *Amanita* species are restricted to sect. *Phalloideae*, and the basal most species do not contain the toxins [16]. Basal amanitin-producing species are therefore valuable for insights into the origin and evolution of the pathway in the genus. *Amanita molliuscula* is just such a species, and several labs have confirmed that its toxin-phenotype is unique as HPLC analysis does not show discernable α-amanitin production (personal communications). Because α-amanitin is present in all other known amanitin-producing mushrooms in Agaricales [4,5], this species would serve as a great model for studying the genetic basis underlining the phenotype. Although genomes of *Amanita* species are becoming more available, expression studies of the amanitin biosynthetic genes are still limited to yes/no level. The first report on the amanitin biosynthetic pathway showed that *AMA1*, an α-amanitin-encoding *MSDIN*, and *PHA1*, a phallacidin-encoding *MSDIN*, are transcribed in mature carpophores [12]. Later, 11 *MSDIN*s were shown to be transcribed in *A. exitialis*, and in total 4 cyclic peptides were found [17]. In *A. bisporigera* and *A. phalloides*, half of the unknown *MSDIN* genes were shown to be transcribed [18]. Our recent report showed over 80% of *MSDIN*s were transcribed in *A. subjunquillea* [19]. The reports so far have indicated that not every *MSDIN* is expressed, so *MSDIN*s are likely not coregulated. Further, it is unknown to what extent the expression level correlates with final cyclic peptide product abundance. Detailed data on the expression of *MSDIN* and other amanitin biosynthetic genes is still lacking.

In this study, *A. molliuscula* was sequenced via PacBio and Illumina RNA-Seq platforms. The *MSDIN* and *POPB* genes were analyzed at both genomic and transcriptomic levels. In addition, cyclic peptide products were analyzed through our genome-guided approach [19], and correlation between expression level and final cyclic peptide products was carefully assessed.

## 2. Materials and Methods

### 2.1. Sample

The mushroom fruiting body used in this study was *A. molliuscula* Qing Cai, Zhu L. Yang & Yang-Yang Cui, a rare species found in China. This particular specimen was collected in Changbai mountains, Jilin Province, Northeast China. The fresh specimen (designated *A. molliuscula* Jilin-China) was identified by Zhu L. Yang. The Latin *molliusculus* refers to the somewhat soft volval limb of the species composed of abundant inflated cells [20]. The type specimen was deposited in the Herbarium of Cryptogams, Kunming Institute of Botany [20], Chinese Academy of Sciences (type, HKAS 77324). The fruiting body was wrapped in aluminum foil on site, immediately put on dry ice, and subsequently stored in liquid nitrogen before use.

### 2.2. Phylogenetic Tree Construction

To confirm the phylogenetic position of *A. molliuscula* among lethal amanitas, a phylogenetic tree was constructed using ITS (internal transcribed spacer) of 12 species from Europe, North America and Asia. These species are *A. bisporigera*, *A. phalloides*, *A. virosa*, *A. molliuscula*, *A. pallidorosea*, *A. subpallidorosea*, *A. rimosa*, *A. subjunquillea*, *A. fuliginea*, *A. exitialis*, *A. franzii* and *A. zangii*. Two of the species, *A. franzii* and *A. zangii*, are nontoxigenic and were used as outgroups. Internal transcribed spacer sequences were obtained from NCBI website (https://www.ncbi.nlm.nih.gov/, accessed on 12 December 2020), and from the sequenced genome in this study. The gene accession numbers and sequence information are listed in Appendix A. Sequences were aligned by MAFFT v7.304b [21] with default settings, and then manually adjusted with BioEdit [22]. Maximum Likelihood analyses and bootstrapping (1000 replicates) were performed using RAxML v7 [23]. For Bayesian inference analyses, MrBayes v3.2.6 [24] was used under the optimal substitution model obtained from MrModeltest [25].

### 2.3. Genome Sequencing and Assembly

The genome of *A. molliuscula* was sequenced and assembled by NextOmics Biosciences, Wuhan, China. The specific process was as follows. Genomic DNA was extracted by Genomic DNA extraction kit (QIAGEN, Germantown, MD, USA) according to the standard operating procedure provided by the manufacturer. The genomic DNA was then sheared by g-TUBEs (Covaris, Woburn, MA, USA) according to the expected size (20 kb) of the fragments for the library. The fragmented DNA of target size was enriched and purified by MegBeads (GenScript, Piscataway, NJ, USA). Next, the fragmented DNA was repaired for the damage and then end-repaired. A 20-kb library was then constructed using a PacBio template prep kit and analyzed with Agilent 2100 Bioanalyzer for quality control. After completion of the library, DNA-enzyme mixture was transferred to PacBio Sequel platform for real-time molecular sequencing. Illumina HiSeq X10 platform was used for error correction. The reads were assembled by Falcon (https://github.com/PacificBiosciences/FALCON-integrate, accessed on 25 November 2018), and the completeness of the gene repertoires was assessed by Benchmarking Universal Single-Copy Orthologs (BUSCO) [26].

### 2.4. Transcriptome Sequencing and Analysis

*Amanita molliuscula* is a species forming mycorrhiza with trees and has not, to date, been cultured artificially. Only one mushroom fruiting body was obtained in this study, and three RNA extractions were made from this sample. RNA concentration was measured using Qubit RNA Assay Kit (Thermo Fisher Scientific, Waltham, MA, USA) in Qubit 2.0 Flurometer (Life Technologies, Carlsbad, CA, USA). RNA integrity was assessed using RNA Nano 6000 Assay Kit (Agilent Technologies, Santa Clara, CA, USA) with the Bioanalyzer 2100 system (Agilent Technologies, Santa Clara, CA, USA). The mRNA was purified from total RNA using poly-T oligo-attached magnetic beads. First strand cDNA was synthesized using random hexamer primer and M-MuLV Reverse Transcriptase (New England BioLabs, Ipswich, MA, USA). Second strand cDNA synthesis was subsequently performed using DNA Polymerase I (New England BioLabs, Ipswich, MA, USA) and RNase H (New England BioLabs, Ipswich, MA, USA). Library quality was assessed on the Agilent Bioanalyzer 2100 system, and transcriptome sequencing performed on Illumina RNA-Seq platform. The quality control of the raw data was carried out with FastP software under default settings [27]. Hisat2 [28] was used to align transcriptome reads and to generate Sam format file, and SAMtools [29] adopted to convert large Sam files into binary BAM format file. The Subread software was used to calculate the read counts [30], and the DESeq2 was subsequently adopted to analyze the differential gene expression of the three transcriptomes [31]. Finally, the assemblies and FPKM values (Fragments Per Kilobase of transcript per Million fragments mapped) were obtained using StringTie with default settings [32].

### 2.5. Venn Diagram Construction

In order to show orthologous gene groups in six lethal amanitas (*A. molliuscula*, *A. bisporigera*, *A. phalloides*, *A. exitialis*, *A. pallidorosea* and *A. subjunquillea*), OrthoFinder [33] was used with default program settings. The outputs were uploaded to JVENN (http://jvenn.toulouse.inra.fr/app/example.html, accessed on 5 March 2020) for diagram construction.

### 2.6. Visualization of the Genome

Visualization software Circos [34] was chosen to map the genome and amanitin biosynthetic genes. The information tracks included location and size of each contig, GC content, gene transcription (FPKM values), distribution of amanitin biosynthetic genes, prediction of genes, transposons and synteny analysis. The GC content was calculated by our in-house scripts. The gene transcription was indicated by FPKM values, and the coordinate information of amanitin biosynthetic genes was obtained by BLAST (NCBI BLAST+ 2.6.0). Synteny was calculated by blastn module. The information was written into seven tracks and loaded into Circos for a genomic overview of *A. molliuscula*.

The synteny analysis of three species producing lethal toxins (*A. molliuscula*, *A. pallidorosea* and *A. exitialis*) was conducted via SyMAP (Synteny Mapping and Analysis Program) 5.0.6 [35]. All alignments were computed using the default parameters. All scaffolds or contigs for all the genomes were loaded in the program.

### 2.7. LC-MS and LC-MS/MS Analyses for Novel Cyclic Peptides

For cyclic peptide extraction, 0.05 g dry weight sample was ground in liquid nitrogen. Then 2 mL buffer was added, using methanol: water: 0.01 M hydrochloric acid (5:4:1) [36,37]. The suspension was transferred to centrifuge tubes, and then heated in a water bath at 75 ℃ for 15 min, followed by centrifugation (12,000 rpm) for 5 min. Finally, the supernatant was filtered with a 0.22 µm polyethersulfone syringe filter (BS-PES-22, Biosharp).

Liquid chromatography–mass spectrometry (LC-MS) and liquid chromatography–tandem mass spectrometry (LC-MS/MS) analyses were conducted through Agilent 1290 Infinity II HPLC system with column (4.6 × 100 mm I.D., particle size 2.7 μm, Agilent Technologies). To detect potential cyclic peptides of *A. molliuscula*, the filtered supernatant was applied on Agilent 6540 UHD precision mass quadrupole time-of-flight (Q-TOF) LC/MS (Agilent Technologies, Santa Clara, CA, USA). The eluent was monitored in positive electrospray ionization (ESI) mode with the capillary voltage at 3.5 kV. The drying gas (N_2_) temperature was 350 °C, and the flow rate was 8 L/min. The mass scan range was 500–1700 m/z. For collision energy in the subsequent MS/MS analysis, a range of 10–70 eV was applied.

For the prediction of monoisotopic masses and molecular formula of potential novel cyclic peptides based on *MSDIN* sequences, the online calculation tool NCBI MIDAs (https://www.ncbi.nlm.nih.gov/CBBresearch/Yu/midas/index.html, accessed on 12 October 2019) was adopted. Then, iminium ions, y-type fragmentation and neutral loss ions were determined based on core peptides of *MSDIN*s using Molecular Weight Calculator v6.50 (https://omics.pnl.gov/software/molecular-weight-calculator, accessed on 5 July 2019). All amino acid composition and manual MS/MS analysis for novel cyclic peptides were based on our genome-guided method [19].

### 2.8. Cloning of MSDIN and POPB Genes in A. molliuscula

Nucleotide sequences of *MSDIN*s and *POPB* genes from the *A. molliuscula* genome were obtained by BLAST (NCBI BLAST+ 2.6.0). Query *MSDIN* and *POPB* sequences came from *A. bisporigera* and *Galerina marginata* [15,18], and the detailed search strategy can be found in our recent report [19]. The nucleotide sequences of *MSDIN*s and *POPB* genes of *A. molliuscula* were aligned with those of *A. bisporigera* by MegAlign v7.1.0. Coding sequences (CDS) of *MSDIN*s and *AmPOPB* (*Am* are the initials of the species and are applied throughout the report) were obtained by reverse transcription PCR (RT-PCR) using primers based on the genomic data. First and second strands of cDNA were obtained using the methods mentioned above. In the genome of *A. molliuscula*, there were three *MSDIN*s encoding α-amanitin. The corresponding amino acid sequences were MSDINATRLPIWGIGCNPCVGDDVTTLLTRGEALC and MSDINATRLAIWGIGCNPCVGDDVTALLTRGEALC (two identical sequences), which were named AmAMA1, AmAMA2-1 and AmAMA2-2, respectively. And the gene encoding β-amanitin was named *AmAMA3.* Two novel cyclic peptides, CylK1 (named *AmCylK1*) and CylK2 (named *AmCylK2*), were discovered in this study (see results). The primers are shown in Appendix A. The structures of the genes were illustrated using the coding sequences (CDS) and genomic DNA sequences on Splign website (https://www.ncbi.nlm.nih.gov/sutils/splign/splign.cgi, accessed on 16 June 2020).

## 3. Results

### 3.1. Phylogenetic Position of A. molliuscula

The genus *Amanita* contains three major clades and 11 subclades [38], among which sect. *Phalloideae* possesses three subclades and one of them includes all known lethal amanitas [38,39]. In order to illustrate the phylogenetic position of *A. molliuscula* among lethal amanitas, 12 species distributed in Europe, North America and Asia were selected for reconstruction. The resultant phylogenetic tree was constructed with 33 ITS sequences, including two nontoxic outgroup species (Figure 1a). Among these species, *A. molliuscula* (shaded in red), including *A. molliuscula* Jilin-China used in this study (Figure 1b), was at the basal position of lethal amanitas, and the branch had robust support (ML bootstrap value = 95, Bayesian posterior probability = 1). The result is consistent with previous studies [20].

### 3.2. Genome and Transcriptome of A. molliuscula

The whole genome shotgun assembly of *A. molliuscula* has been deposited at DDBJ/ENA/GenBank as part of the batch accession JAEBUT000000000. The assembly results are shown in Figure 2. The amount of clean data was 4.46 Gb, the assembled genome was 71.17 Mbp and the sequencing depth was about 60 x. BUSCO prediction with Basidiomycota settings implied that 87.3% of the core genes were found. Transposable elements (TEs) accounted for 64.78% of the genome.

Because *A. molliuscula* is difficult to encounter in the field and cannot be cultured artificially, we used three technical replicates in the transcriptome sequencing. Differential expression analysis on the transcriptomes showed that the three replicates were highly similar, and there was no differential expression in all of the amanitin biosynthetic genes across the three replicates (Appendix A). The result strongly suggested that the replicates were consistent, and one of the three transcriptomes was chosen for display in the following results.

The genome and transcriptome features were visualized as a circular diagram using Circos software (Figure 2). The tracks (from the outer ring) represented contigs, GC content, transposons, distribution of toxin biosynthetic genes, transcriptional gene expression (FPKM value), prediction of genes and synteny. In total, 17 *MSDIN* genes (21 sequences) were discovered with 14 being novel, and they scattered among contigs with some degree of clustering. The *AmPOPB* gene was located on an isolated contig. Colors of track **V** represent expression levels of the predicted genes, and the same color scheme was applied to all other expression results in this report. Expression levels of amanitin biosynthetic genes (*AmPOPB* and *MSDIN*s) and housekeeping genes (*AmPOPA* and *rbp2*) were calculated, and the results shown in Table 1 and Figure 2 (color coded). The transcription of *MSDIN* genes varied greatly, with FPKM ranging from 0 to 357.26. The expression of most *MSDIN* genes was low with FPKM values below 1 (black and green). Among them, the FPKM values of four genes with core peptides of LIFLPPFIPP, FFIIFFIPP, WFFFFYP and FNILPLLLPP, were 0. The *MSDIN* genes encoding major toxins were largely low. The α-amanitin-encoding gene had three copies (*AmAMA1*, *AmAMA2-1* and *AmAMA2-2*) with FPKM values of 0.24, 0.15 and 2.7, respectively. The FPKM value of the β-amanitin-encoding gene (*AmAMA3*) was 0.78. Two *MSDIN* genes (*AmCylK1* and *AmCylK2*) with core peptides GFGFIP and GKVNPP were highly expressed, and the FPKM values were 357.26 and 113.45, respectively. Four *MSDIN*s were transcribed with FPKMs between 1 and 60.

The FPKM value for *AmPOPB*, a key functional gene for toxin synthesis, was 178.43, 25 times higher than that of *AmPOPA*, a housekeeping gene also belonging to the *POP* gene family. The value was also much higher than that of the other housekeeping gene *rbp2*. The results showed that the cyclic peptide biosynthetic pathway was highly active in *A. molliuscula*.

### 3.3. Orthology and Synteny Analysis of A. molliuscula

The orthology analysis of *A. molliuscula*, *A. bisporigera*, *A. phalloides*, *A. exitialis*, *A. pallidorosea* and *A. subjunquillea* is shown in Figure 3a. In total, 10,513 orthogroups were included. There were 1860 common orthogroups among the six species. *Amanita bisporigera* had 2158 species-specific orthogroups, which was the largest of the six species. The least was in *A. phalloides*, which had 819 unique orthogroups.

In order to compare the whole genome of *A. molliuscula* with other lethal species, we selected *A. exitialis*, the closest lethal species to *A. molliuscula*, and *A. pallidorosea*, one of the most distant species, for synteny analysis. It was obvious that *A. molliuscula* shared more synteny with *A. exitialis* than with *A. pallidorosea* (Figure 3b), which was consistent with their phylogenetic relationships.

### 3.4. Amatoxins and Novel Cyclic Peptides from A. molliuscula

For cyclic peptide analyses, extracts of *A. molliuscula* were analyzed by HPLC, LC-MS and LC-MS/MS. HPLC analysis confirmed that only β-amanitin was discernable in *A. molliuscula*. Following our genome-guided approach [19], we used mass spectroscopy to further investigate other potential cyclic peptides based on theoretical molecular masses of potential cyclic peptides encoded by the 17 *MSDIN* sequences (Table 2). β-Amanitin was readily detected, and surprisingly, a trace amount of α-amanitin was also discovered. In addition, phallotoxin was not detected in this species.

α-Amanitin has a formula of C_39_H_54_N_10_O_14_S and the measured mass of the [M + H]^+^ ion is 919.3616, with mass discrepancy of 0.22 ppm (Figure 4a). The formula of β-amanitin is C_39_H_53_N_9_O_15_S and the measured mass of the [M + H]^+^ ion is 920.3465, with mass discrepancy of 1.09 ppm (Figure 4b). In addition, masses of two potential cyclic peptides were found. They match the core regions of two MSDIN sequences, i.e., MSNINALRLPGFGFIPYASGDVDYTLTRGESLS (candidate novel cyclic peptide named CylK1) and MSDINATRFPGKVNPPYVGDDVDDIIIRGEKLC (candidate novel cyclic peptide named CylK2), respectively. The core peptides were underlined. The measured molecular masses of CylK1 and CylK2 were 619.3244 and 593.3411, respectively (Figure 4c,d). Both mass discrepancies were within 1 ppm (Table 2).

MS/MS fragments of CylK1 and CylK2 underwent further manual analyses. For CylK1, iminium ions for Ile and Phe were found, and the measured daughter ions were 86.0967 and 120.0808, respectively (Figure 4e). Then, 7 y-type ions were confirmed according to the core peptide sequence, i.e., GP, GF, GGI, GFP, GFPIG, GFPIF and GFPFG. Based on the result, CylK1 was determined to be a novel cyclic peptide.

Similar procedures were applied to CylK2 (Figure 4f). Eleven fragments were matched with predicted ions based on the core peptide sequence. First, Pro was found by iminium ions, with the daughter ion mass at 70.0654. Then, the y-type ions, GP, GK, GKP, KPP, GPPN, GKPP and GKPPN, were matched with fragmentation ions. In addition, loss of neutral ion (NH_3_) was detected on three fragments in MS/MS, and the following analyses showed that it was due to the loss of NH_3_ on Lys. Collectively, the results strongly suggested CylK2 to be a novel cyclic peptide.

### 3.5. Structures of AmPOPB, AmAMA1, AmAMA2, AmAMA3, AmCylK1 and AmCylK2

The *MSDIN* genes of *A. molliuscula* encode no phallotoxins (consistent with the MS result) but two known amatoxins, i.e., α-amanitin and β-amanitin. There were three sequences encoding α-amanitin (*AmAMA1*, *AmAMA2-1* and *AmAMA2-2*). *AmAMA1* and *AmAMA2* differed in the leader peptide region (MSDINATRLP vs. MSDINATRLA) and in recognition sequences (CVGDDVTTLLTRGEALC vs. CVGDDVTALLTRGEALC). The last amino acid of *AmAMA2* in the leader peptide region was Ala residue. There was only one sequence encoding β-amanitin, and the last amino acid in the leader peptide region of this sequence was also Ala residue. In contrast, this particular site in all other MSDIN sequences is Pro residue [12,13,14,15,17,40,41]. Both *AmCylK1* and *AmCylK2* coded for 6-aa peptides.

The gene structure of the above five *MSDIN* genes, which have corresponding cyclic peptide products, and the structure of *AmPOPB*, were predicted on Splign website and confirmed by cloning. All five *MSDIN* genes contain four exons and three introns each, similar to those from other lethal *Amanita* species. *AmPOPB* contained 18 exons and 17 introns and encoded 731 amino acids. Nucleotide sequences, amino acid sequences and structures for these genes are displayed in Appendix A.

## 4. Discussion

Amanitin-producing *Amanita* species are restricted in the sect. *Phalloideae*, and only the basalmost species in the section do not contain the cyclic peptide toxins [16]. *Amanita molliuscula* is a basal species found in the lethal *Amanita* clade and no cases of poisoning have been reported with this species. This mushroom could offer insights into the original form of the amanitin biosynthetic pathway in this genus. This study showed that phallotoxins were not present in the species, which sets it apart from all other known lethal *Amanita* species. In addition, α-amanitin was detected only at trace level, indicating this species is less toxic compared with its allies. In total, 21 genes encoding 17 *MSDIN*s were discovered in *A. molliuscula*, with 14 *MSDIN*s encoding potential novel cyclic peptides. In addition to 3 *MSDIN*s coding for α-amanitin and one for β-amanitin, the core peptide LNILPFHLPP has been reported in *A. pallidorosea* [13]. Two novel cyclic peptides (CylK1 and CylK2) were discovered, which are the smallest *MSDIN*-related cyclic peptides and share no similarity to the others in the group.

All previously known lethal *Amanita* species produce both amatoxins(s) and phallotoxins(s). In previous studies, six genomes of lethal amanita have been published, i.e., *A. bisporigera*, *A. phalloides*, *A. subjunquillea*, *A. pallidorosea*, *A. exitialis* and *A. rimosa* [12,18,19]. These species contain at least three major toxins, while *A. molliuscula* possesses only two amatoxins, i.e., α-amanitin (trace amount) and β-amanitin. Given that it is a basal species of lethal *Amanita*, this may indicate that the origin of amatoxins predates that of phallotoxins. There are only 17 unduplicated *MSDIN* sequences in *A. molliuscula*, showing the least expansion of the *MSDIN* family (31,33,24,29,23,29) within *Amanita*. Moreover, the last amino acid residue of the leader peptide in *AmAMA2* and *AmAMA3* is Ala, while all other *MSDIN*s possess a highly conserved Pro. Collectively, we speculate that many of the above unique characteristics are linked to the fact that this particular species is at the basal position of lethal *Amanita* species.

Similar to our previous study [19], a majority of the *MSDIN* genes are expressed at the transcription level in *A. molliuscula*. Some *MSDIN*s and *AmPOPB* are highly expressed (compared with housekeeping genes), suggesting that the amanitin biosynthetic pathway is highly active at the mature fruiting-body stage. Further, highly differential expression was found in *MSDIN* genes, indicating that cyclic peptides are not equally expressed. Our preliminary expression data (without technical replicates) on *A. subjunquillea* and *A. exitialis* showed very similar patterns (data not shown). Thus, in the mature fruiting body of *A. molliuscula*, only a fraction of cyclic peptides are actively synthesized, while most, including the major toxins, are suppressed. It is common knowledge that amanitins are present at all stages of fruiting body development [40,41], indicating these toxins are biosynthesized at early stage(s). Consistently, previous studies on the expression of α-amanitin and β-amanitin in *A. exitialis* showed that the expression of these genes is higher at the immature stage of the fruiting body [17]. The accumulation of these toxins could inhibit their expression at later stages. Regarding *AmPOPB* expression, the key enzyme of the pathway, the high expression strongly suggests that the metabolic pathway is actively producing cyclic peptides. The production probably includes CylK1 and CylK2 since the highest transcription was associated with the two underlying *MSDIN*s. The shift of the peptide production may be linked to the functions of these peptides, which may be needed at different developmental stages. In conclusion, there is a clear correlation between transcription level and final cyclic peptide products, as highly expressed *MSDIN*s tend to have detectable peptide products. However, low expression of *MSDIN* genes may not correlate with peptide accumulation, such as the major toxins, which indicates a sequential biosynthesis of the peptides under a sophisticated control mechanism.

The genome size of *A. molliuscula* is slightly larger than other sequenced *Amanita* species. One likely reason is that this species contains higher percentage of transposable elements (64.78%). *AmPOPB* is not clustered with any of the *MSDIN* genes, which is consistent with previous findings [14,18]. *MSDIN* genes scatter across the genome, but the distribution is not entirely random. Some *MSDIN*s tend to group together (linked or close to one another), showing some degree of clustering, while others can be totally isolated by themselves (Figure 2). Based on the transcriptome, grouped *MSDIN*s does not share comparable expression levels; instead, they exhibit differential patterns, showing that even *MSDIN*s in close proximity are not coregulated. *AmPOPA*, although not an amanitin biosynthetic gene, was also checked for clustering, and it is not located near any of the biosynthetic genes. Overall, our results confirmed that amanitin biosynthetic pathway does not assume the form of a classic gene cluster, which is consistent with other reports [14,18].

Cyclic peptides produced in *Amanita* species, such as α-amanitin and phalloidin, are valuable natural products [6,7,8,9], and only very recently, chemical synthesis of α-amanitin is achieved [42,43,44]. The two novel cyclic peptides (CylK1 and CylK2) discovered in this study have unique structures. Not only are they the first 6-aa cyclic peptides found in amanitin-producing mushrooms, they share no obvious sequence similarity to other cyclic peptides. Before this report, the shortest cyclic peptides comprised 7 amino acids, such as the phallotoxins and virotoxins [1,5]. There are also longer peptides containing up to 10 amino acids, such as antamanide and related peptides [1,5,19]. Our recent report showed a cyclic peptide containing 11 amino acids, to date the longest [19]. Both CylK1 (Cyclo(GFGFIP)) and CylK2 (Cyclo(GKVNPP)) have no further posttranslational modifications. They lack Trp and Cys, and therefore no tryptathionine bridge [45] could occur. CylK2 has an alkaline amino acid residue (Lys) with double Pro. In comparison, CylK1 is much more hydrophobic and only has one Pro residue. CylK1 and CylK2 do not share obvious sequence similarities with each other or with any other cyclic peptides, and the novelty of the structure may indicate novel functions. We are synthesizing the peptides for further studies.

## 5. Conclusions

A basal species of amanitin-producing *Amanita* species, *A. molliuscula*, was sequenced, yielding 14 novel *MSDIN* genes. Global expression of amanitin biosynthetic genes was illustrated in the species. At mature fruiting body stage, the pathway is highly active with differential expression of amanitin biosynthetic genes. Led by expression data, two novel cyclic peptides which have novel structure and share no similarity to known cyclic peptides were discovered through a genome-guided approach. The result indicates that *MSDIN* expression level is associated with final peptide production. Cyclic peptides in *A. molliuscula* are sequentially expressed; major toxins are not actively expressed at the mature fruiting body stage. This research offers a first in depth look into the expression of amanitin biosynthetic genes.

## Figures and Tables

**Figure 1 jof-07-00384-f001:**
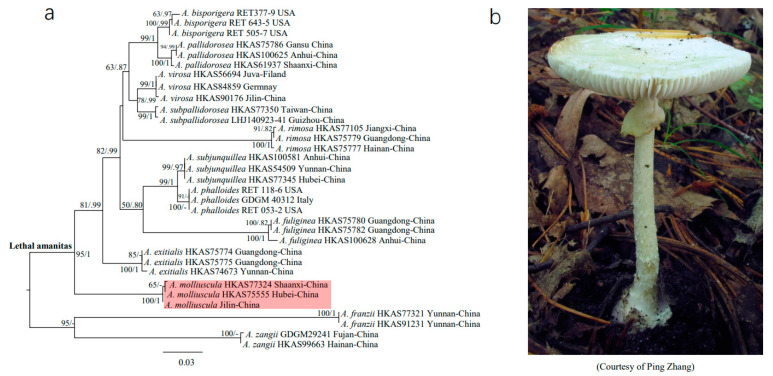
(**a**) Phylogenetic tree of lethal amanitas. (**b**) *Amanita molliuscula* in the field.

**Figure 2 jof-07-00384-f002:**
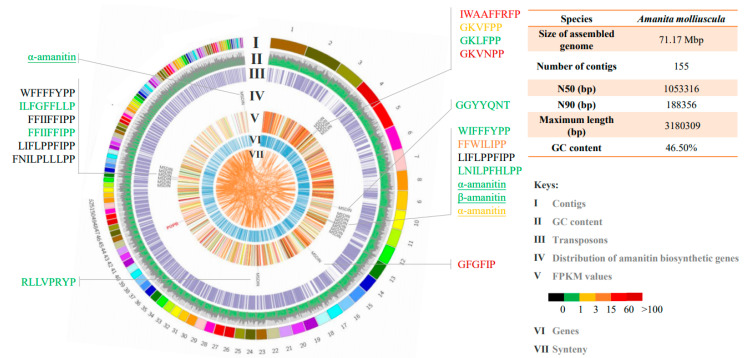
Genome of *Amanita molliuscula* and transcription of the amanitin biosynthetic genes. Tracks: **I**, Contigs; **II**, GC content; **III**, Transposons; **Ⅳ**, Distribution of amanitin biosynthetic genes; **Ⅴ**, FPKM values of genes; **Ⅵ**, Distribution of genes; **Ⅶ**, Synteny. Statistics of the genome is shown on the upper right. Amanitin biosynthetic genes are shown in Track **IV**, with expression levels of *MSDIN* genes shown surrounding the genome circle (keys on the lower right).

**Figure 3 jof-07-00384-f003:**
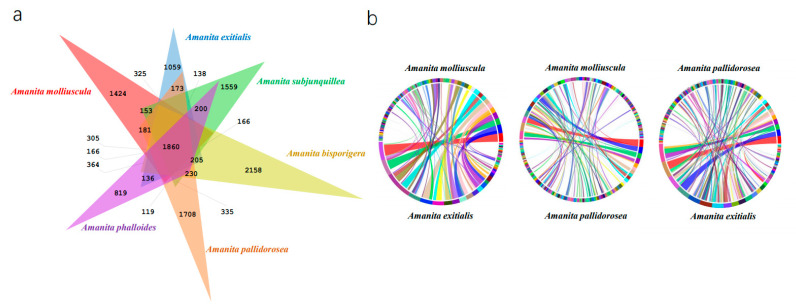
(**a**) Venn diagram of predicted orthogroups of amanitas in North America, Europe and Asia. (**b**) The synteny analysis of *Amanita molliuscula*, *A. pallidorosea* and *A. exitialis*.

**Figure 4 jof-07-00384-f004:**
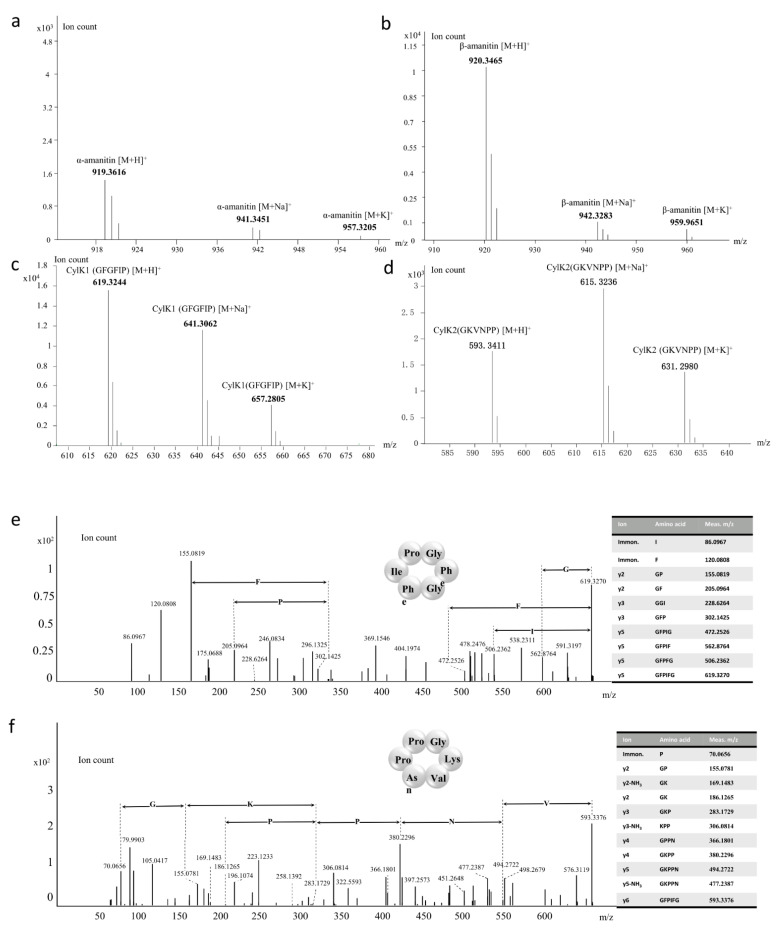
Mass spectrometry spectrograms of known and novel cyclic peptides produced in *Amanita molliuscula* ((**a**): α-amanitin, (**b**): β-amanitin, (**c**): CylK1 (Cyclo(GFGFIP)), (**d**): CylK2 (Cyclo (GKVNPP)). Manual analysis of MS/MS fragmentation of the two novel cyclic peptides, CylK1 (**e**) and CylK2 (**f**).

**Table 1 jof-07-00384-t001:** Transcription of amanitin biosynthetic and housekeeping genes of *Amanita molliuscula.*

	Peptides and Proteins			FPKM Values	Notes
Amanitin biosynthetic genes	MSDINATRLP	IWGIGCNP	CVGDDVTTLLTRGEALC	0.24	α-amanitin 3 copies
	MSDINATRLA	IWGIGCNP	CVGDDVTALLTRGEALC	0.15	α-amanitin
	MSDINATRLA	IWGIGCNP	CVGDDVTALLTRGEALC	2.70	α-amanitin
	MSDINATRLA	IWGIGCDP	CVGDDVTALLTRGEALC	0.78	β-amanitin
	MSNINALRLP	GFGFIP	YASGDVDYTLTRGESLS	357.26	CylK1
	MSDINATRFP	GKVNPP	YVGDDVDDIIIRGEKLC	113.45	CylK2
	MSNINASRLP	IWAAFFRFP	CVGDEVDGILRSGESLC	38.94	
	MTDINATRLP	FFWILIPP	CVDDVDNTVHSGDNLC	4.63	
	MSDINATRFP	GKVFPP	YVGDDVDDIIIRGEK	2.07	
	MSNINATRFP	GKLFPP	YVGDDVDDIIIRGDKLC	0.93	
	MSDINASRLP	RLLVPRYP	CIDEDAEAILRSGECL	0.90	
	MTDINATRLP	ILFGFFLLP	CVDGVDNTLHSGENLC	0.66	
	MSDINSIHLP	GGYYQNT	FVGDDVEGILNRGERLC	0.61	
	MTDINATRLP	LNILPFHLPP	CVDDVDNTLHSGENLC	0.29	
	MTDINATRLP	FFIIFFIPP	CVDDVDNTLHSGENLC	0.25	2 copies
	MTDINATRLP	FFIIFFIPP	CVDDVDNTLHSGENLC	0.00	
	MTDINATRLP	WIFFFYPP	CVDDVDNTLHSGENLC	0.09	
	MTDINATRLP	LIFLPPFIPP	CVDDVDNTLHSGENLC	0.00	2 copies
	MTDINATRLP	LIFLPPFIPP	CVDDVDNTLHSGENLC	0.00	
	MTDINATRLP	WFFFFYPP	CVDDVDNTLHSGENLC	0.00	
	MTDINATRLP	FNILPLLLPP	CVDDVDNTLHSGENLC	0.00	
	*POPB*			178.43	
Housekeeping genes	*POPA*			6.99	
	*rbp2*			46.96	

**Table 2 jof-07-00384-t002:** Theoretical and measured molecular masses of cyclic peptides in *Amanita molliuscula.*

Cyclic Peptides	Elemental Composition	Theoretical (m/z)	Measured (m/z)	δ (ppm)
IWGIGCNP (α-amanitin)	C_39_H_54_N_10_O_14_S + H^+^	919.3614	919.3616	0.2175
IWGIGCDP (β-amanitin)	C_39_H_53_N_9_O_15_S + H^+^	920.3455	920.3465	1.0865
GFGFIP (CylK1)	C_33_H_42_N_6_O_6_ + H^+^	619.3239	619.3244	0.8073
GKVNPP (CylK2)	C_27_H_44_N_8_O_7_ + H^+^	593.3406	593.3411	0.8427
IWAAFFRFP	C_61_H_77_N_13_O_9_ + H^+^	1136.604	N/A	N/A
GGYYQNT	C_35_H_45_N_9_O_12_ + H^+^	784.326	N/A	N/A
RLLVPRYP	C_48_H_78_N_14_O_9_ + H^+^	995.6149	N/A	N/A
LNILPFHLPP	C_58_H_87_N_13_O_11_ + H^+^	1142.6721	N/A	N/A
LIFLPPFIPP	C_62_H_90_N_10_O_10_ + H^+^	1135.6914	N/A	N/A
FFWILIPP	C_57_H_75_N_9_O8 + H^+^	1014.5811	N/A	N/A
FFIIFFIPP	C_64_H_83_N_9_O_9_ + H^+^	1122.6387	N/A	N/A
WIFFFYPP	C_63_H_71_N_9_O_9_ + H^+^	1098.5448	N/A	N/A
GKVFPP	C_32_H_47_N_7_O_6_ + H^+^	626.3661	N/A	N/A
GKLFPP	C_33_H_49_N_7_O_6_ + H^+^	640.3817	N/A	N/A
ILFGFFLLP	C_58_H_81_N_9_O_9_ + H^+^	1048.623	N/A	N/A
WFFFFYPP	C_66_H_69_N_9_O_9_ + H^+^	1132.5291	N/A	N/A
FNILPLLLPP	C_58_H_91_N_11_O_11_ + H^+^	1118.6972	N/A	N/A

## Data Availability

The Whole Genome Shotgun assembly was deposited at DDBJ/ENA/GenBank as part of the accession JAE-BUT000000000.

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
