# Peer review of "Differential Expression of Amanitin Biosynthetic Genes and Novel Cyclic Peptides in Amanita molliuscula"

_jof, 2021, doi:10.3390/jof7050384_

Round 1

Reviewer 1 Report

Comment 1: Please change the title.

Comment 2: Ln. 22. Write abbreviation for POPB.

Comment 3: There is a lot of information in the results section, but did find the detailed discussion. Rewrite the discussion part.

Comment 4: Manuscript is not concluded. Please conclude clearly.

Comment 5: Try to include more recent references for discussion and introudction.

Reviewer 2 Report

The work presented by Lüli et al. studies the expression of amanitin biosynthetic genes and novel cyclic peptides in Amanita molliuscula. To this end, transcriptomic and genomic analyses were performed of a specimen of this species. These analyses revealed the presence of several genes belonging to the MSDIN family including two novel cyclic peptides with different expression patterns. The study overall is very interesting and well presented. I recommend accepting the manuscript after the following revisions.

Specific comments

Line 2: Please, remove “Highly” from the title.

Lines 70-71:"expressed through the transcriptome" is redundant, please replace it by "expressed in A. exitialis" or "transcribed in A. exitialis".

Lines 72-73: "expressed at the transcription level" is redundant, please replace it by "expressed" or "transcribed".

Line 81: Please, check this reference.

Lines 85-92: Please, provide more details on the identification of this specimen, particularly the nomenclature followed.

Lines 110-111: Please, mention the method followed for the DNA extraction.

Line 122: Please, mention the method followed for the RNA extraction.

Line 123: Please, explain how the cDNA was obtained.

Line 144: Please, check this "on scripts".

Line 176: Please provide more details on the cloning method and the PCR amplification procedures. Primer sequences should go to a table within the supplementary material.

Lines 229-233: Please, show the results of that analysis.

Lines 237-238: Please, provide more details on how the annotation of these genes was accomplished in the Materials and Methods section.

Line 246: Please, check this name (FNAIL...) here and in the figure 2.

Line 259, Figure 2: This figure should be bigger.

Lines 250-251: These two genes were CylK1 and CylK2? Please clarify it in the text.  

Line 263, Table 1: The “Genes” column is confusing as most of the information provided correspond to peptide sequences, not genes. Please, clarify this point.

Line 327: Which five MSDIN genes? CylK1 and CylK2 were not mentioned in the previous paragraph. Please, clarify it.

Reviewer 3 Report

Dear Editor,

Thanks for the opportunity to review a manuscript titled "Highly differential expression of amanitin biosynthetic genes and novel cyclic peptides in Amanita molliuscula" by Luli et al.

The manuscript is generally well-written and provides insights into amanitin biosynthetic pathway in Amanita molliuscula.

However, the whole text needs polishing, preferentially by a native English speaker.

Abstract:

  1. Needs some polishing.

Introduction

  1. Add a study hypothesis and/or aim.
  2. Polish the section

Materials and Methods

  1. "High quality genomic DNA " is somewhat vague (add information about the extraction kit and how quality was measured).
  2. Manufacturer information should be added throughout the section.
  3. A statistical section is missing.

Results

  1. Polish the section

Discussion

  1. Please include references to back important claims throughout the section, e.g., "It is common knowledge that amanitins are present at all stages of fruiting body development, indicating these toxins are biosynthesized at early stage(s)"
  2. A conclusion should be added at the end of the section.
  3. Polishing is needed.

Reviewer 4 Report

"The fruiting body was wrapped in tin foil on-site". Perhaps the authors used aluminum foil? 

The structure assignment of the two novel peptides seems to be somehow tentative. This should be mentioned accordingly. It would be nice to have more analytical and other data on these peptides. Please show the structure of the cyclic peptide as a structural formula.

I am confused about the sequence of CylK1+2. In the supplement, relatively long sequences are given, in Fig. 4 hexapeptides are shown.

I understand that this paper seems to be focused on the gene clusters responsible for the synthesis of the respective peptides. However, some information of about the detailled function of the expressed and non-expressed enzymes would be helpful. Is there any rational explanation for the formation of the novel peptides based on the expression pattern of enzymes?

On the other hand, the potential discovery of two new peptides should be shown more prominently. I assume that this is an important outcome of the study.

Supplementary file 1 and Figure S1 seem to be lacking.

The two novel cyclic peptides (CylK1 and CylK2): Are they toxic, might they have a biological function, or are they defective products due to the lack of some enzymes?
